# Animal Welfare Guidelines for International Development Organisations in the Global South

**DOI:** 10.3390/ani14132012

**Published:** 2024-07-08

**Authors:** Paul Ssuna, Andrew Crump, Karin Siegmund

**Affiliations:** 1School of Veterinary Medicine and Animal Resources, Animal Welfare Competence Center for Africa (AWeCCA), Makerere University, Kampala P.O. Box 7062, Uganda; 2Animal Welfare Science and Ethics, Royal Veterinary College, London NW1 0TU, UK; 3Centre for Philosophy of Natural and Social Science, London School of Economics and Political Science, London WC2A 2AE, UK; 4Welttierschutzstiftung, 10117 Berlin, Germany; ks@welttierschutz.org

**Keywords:** animal welfare, charity, government, international development organisation, NGO

## Abstract

**Simple Summary:**

International development organisations have improved billions of human lives in the Global South, but the welfare of animals is not usually “on their radar”. This is problematic because their activities often affect animals, either directly (e.g., livestock in their own projects) or indirectly (e.g., advising governments on agriculture policy). Poor welfare standards risk organisations’ reputations, particularly with donors; they reduce livestock lifespans and productivity, harming recipients; and they cause animals unnecessary pain and suffering. These guidelines were developed through extensive stakeholder engagement with organisations, donors, and recipients, especially in Africa. In these guidelines, we lay out the basic principles of good animal welfare. This includes sections on appropriate food, housing, veterinary care, transport, and slaughter. We hope that, by adopting these guidelines, international development organisations will promote good animal welfare.

**Abstract:**

International development organisations have improved billions of human lives in the Global South. However, in both their projects and advice to governments, most of these organisations neglect animal welfare. This blindspot matters. Poor welfare standards risk the organisation’s reputation, particularly with donors; they reduce livestock lifespans and productivity, harming recipients; and they cause animals unnecessary pain and suffering. Here, we set out animal welfare guidelines for international development organisations. They were developed through extensive stakeholder engagement with organisations, donors, and recipients, especially in Africa. To comprehensively cover animal welfare, the guidelines encompass governance structure within the organisation, staff training, standard operating procedures, water, food, housing, social isolation, enrichment, drainage and waste disposal, disease, invasive procedures, transport, slaughter, breeds, record-keeping, and monitoring and evaluation of success. We urge international development organisations to adopt and institutionalise these guidelines, so they promote good animal welfare.

## 1. Introduction

International development organisations (IDOs) aim to promote economic growth and address social and environmental issues, especially in the Global South. The term encompasses government and intergovernmental bodies (such as the United Nations Development Programme, World Health Organisation, and World Bank) and non-governmental organisations (such as Save the Children, Oxfam International, and Doctors without Borders). These organisations provide or facilitate healthcare, education, infrastructure, emergency relief, and other services [1], which have improved billions of human lives.

One issue, which IDOs have historically not considered, is animal welfare [2]. Although there is no consensus definition of animal welfare, the term usually covers an animal’s physical health and psychological wellbeing [3]. Historically, the focus was on alleviating negative states, such as hunger, pain, and disease [4]. Welfare scientists now recognise that this approach can, at best, achieve neutral welfare [5]. Good welfare requires opportunities for positive experiences, such as rewarding social interactions and cognitive stimulation [6].

Some international development organisations, especially in the organic farming sphere [7], have taken steps towards ensuring good animal welfare on their projects. For example, to promote animal welfare in international development cooperation, the World Organisation for Animal Health made this the subject of its “Third Global Conference on Animal Welfare” (Malaysia, November 2012) [8]. However, even in 2015, none of these development organisation partners were assessing the potential welfare impacts of their programmes. Many also ignored animal welfare in their policy advice to governments (e.g., when providing technical assistance on livestock policy). For most international development organisations, animal welfare is simply not “on the radar” (although some important counterexamples also exist [9]. 

International development organisations might inadvertently cause animal welfare issues in various ways. As an example, to promote economic development and food security, they may fund or recommend the construction of livestock facilities that cause severe animal welfare issues. This is especially true when operators are not properly trained in how to manage or maintain such facilities. In the Global North, for instance, cattle are increasingly housed indoors all year round [10], despite continuous housing leading to poor welfare outcomes [11,12,13,14]. If international development organisations were to promote continuously housed cattle facilities in the Global South, they should at least consider the associated welfare issues. Other examples of IDOs potentially creating welfare issues are the provision of livestock to recipients without offering training or guidance on appropriate husbandry [15]; funding animals but not other infrastructure necessary to meet their needs (e.g., housing or waste disposal systems); and providing temperate livestock breeds unsuited to tropical climates.

Incorporating animal welfare into the decision-making process nonetheless carries potential costs for international development organisations. Resources are finite, and other needs or interests may compete with animal welfare. Specifically, international development organisations often help the world’s poorest people. In trade-offs between human lives and animal welfare, we may be minded to prioritise humans.

However, there are compelling reasons for international development organisations to consider animal welfare. Fundamentally, minimising animals’ pain and suffering is widely perceived as the “right thing to do” [14,15]. This is a consensus view among the public in both the Global North [16] and the Global South [17]. Around the world, there is often a culture of care, respect, and reverence for animals [18]. Moreover, ensuring good animal welfare (especially good health) can bring economic benefits, as animals live longer and perform better [19]. Major international donors are also moving towards allocating funding based on animal welfare considerations [20]. Effective and transparent animal welfare policies are, therefore, crucial for international development organisations to maintain their reputations, both among beneficiaries and donors.

Good animal welfare can also aid international development. For instance, high-welfare livestock systems are often more productive and command higher prices (economic [21]; they can reduce the risk of transmitting diseases which may infect humans (social [22]); and they often have smaller carbon footprints (environmental) [23]. To our knowledge, however, there is no set of animal welfare guidelines tailored for international development organisations.

Here, we develop animal welfare guidelines for international development organisations operating in the Global South, especially Africa. We first outline our methodology, including extensive stakeholder engagement. The guidelines themselves were designed to strike a balance: not especially restrictive or onerous but, nonetheless, establishing an acceptable baseline of welfare. We also note that, in addition to lacking a welfare policy, animal welfare issues can arise due to an inadequate policy or failures to implement the policy. These guidelines are primarily meant to address a lack of or insufficient policy, but also provide standardised guidelines that facilitate implementation. Our aim is for international development organisations to adopt and institutionalise these guidelines, so they actively foster good animal welfare practice.

## 2. Materials and Methods

We developed the guidelines through extensive engagement with international development organisations in the Global South. This involved a survey, development project visits, and a webinar and a conference presentation to invite feedback. Below, we describe the entire process that culminated in the development of these guidelines (which we will then outline in the rest of this paper).

In March 2021, we released an online survey using Kobo Collect software v2021.2.4 (Kobo Organization, Cambridge, MA, USA). The aim was to learn what, if anything, international development organisations were already doing for animal health and welfare in their projects. We sent the survey to international development organisations that have (or have had) a project involving animals in Africa. To identify these, we were given a list of non-governmental organisations from the Welttierschutzstiftung (WTS). WTS is a German charitable foundation, established in 2015 to promote animal welfare worldwide. Their projects focus on improving animal welfare through academic education and continuous professional development in emerging and developing countries, on scientific research into animal welfare, and on raising awareness about animal welfare in society. We also contacted organisations that have participated in WTS’s animal welfare webinars. We emailed the survey to 117 potential participants. Although we only received 15 complete responses (response rate: 12.82%), the respondents represented international development organisations from North, East, South, West, and Central Africa.

Although the survey revealed gaps in animal welfare knowledge and practice, it was unclear whether the results accurately represented the situation “on the ground”. One of the authors, Paul Ssuna, therefore, visited four respondents’ development projects—two in Uganda, one in Rwanda, and one in Kenya. The organisations varied in size. In Uganda, both organisations work at the local level (in three districts). In Kenya and Rwanda, the organisations work across the whole country. These locations were selected for ease of transport. On these project visits, Ssuna (a fully qualified veterinarian) clinically assessed the animals, investigated their housing and environment, and discussed the animals’ care with their owner. The project visits revealed that animal welfare was much worse than the IDOs had indicated in their responses. For example, one IDO claimed to have standard operating procedures (SOPs) for animal husbandry and training for all recipients of animals. There was no evidence of either on the project visit.

The survey and project visits demonstrated that IDOs needed practical animal welfare guidelines. One option was the World Organisation for Animal Health’s (WOAH) existing welfare standards [24]. We, therefore, organised a webinar on Friday, 26 November 2021, where we asked international development organisations if these standards were suitable for development projects. The feedback was overwhelmingly negative. Participants said that the WOAH standards were too general, too complex, and did not reflect on the different situations in different countries. Project managers, they emphasised, are not vets, so any international development organisations’ welfare guidelines must be simpler if they are to be practically implemented. Based on this feedback, we drafted our own animal welfare guidelines, which are tailored to international development organisations.

We drafted our guidelines to complement the existing WOAH guidelines. In particular, they specifically address animal welfare issues during IDO projects (as revealed by surveys and site visits), but not welfare issues at a country level (the stated aim of the WOAH guidelines). For example, survey respondents were keen for us to cover animal housing and feeding, and these arose as issues on project visits. We, therefore, included both as subsections within the guidelines. Although we incorporated aspects of the WOAH standards, we simplified them and focused on identified gaps.

Based on international development organisations’ feedback and our own experience, we also included institutionalisation in the guidelines. The disparity between survey responses and project visits highlighted that international development organisations may have animal welfare policies, but these are not apparent on their projects. Named individuals within and outside international development organisations must, therefore, be responsible for ensuring that guidelines are followed. This is why the guidelines include a steering committee (consisting of internal staff who monitor implementation and outcomes) and an advisory committee (consisting of external stakeholders, such as donors and government agencies, who can independently verify standards).

After completing a draft of the guidelines, we invited feedback from international development organisations. We first shared them with the veterinary boards in Tanzania, Uganda, Nigeria, Botswana, and Ethiopia. Then, in October 2022, we presented the draft guidelines on the first day of the Africa Animal Welfare Conference in Botswana. This event had representatives from over 30 African countries’ Ministries of Agriculture, almost all African NGOs, major animal welfare NGOs from the US and Europe, and academics. Over the next two days, we invited feedback from these diverse stakeholders. We incorporated this feedback into the guidelines and, on the third day, presented the revised guidelines. Seventy-seven delegates signed a petition to approve these revised guidelines, including animal welfare charities in Africa, Europe, and USA, development organisations in Africa, and government ministries concerned with animals. The guidelines outlined in this paper have, therefore, achieved consensus support.

## 3. The Guidelines

### 3.1. General Principles

These guidelines apply to domestic animals commonly used in development projects in the Global South, such as cattle, goats, sheep, pigs, donkeys, camels, and poultry. Recipients may receive these animals to raise for food, sale, or breeding, or they may be used in scientific research. Whatever the context, international development organisations will ensure that animals have the “Five Freedoms” [25], which underpin animal welfare legislation worldwide [26].

Freedom from hunger and thirst, by ready access to fresh water and a diet to maintain full health and vigour.Freedom from discomfort, by providing an appropriate environment including shelter and a comfortable resting area.Freedom from pain, injury, and disease, by prevention or rapid diagnosis and treatment.Freedom to express normal behaviour, by providing sufficient space, proper facilities, and company of the animal’s own kind.Freedom from fear and distress, by ensuring housing and husbandry which minimise suffering.

Although the Five Freedoms helpfully summarise some basic principles of animal welfare, these guidelines necessarily build on them. In particular, the Five Freedoms focus on negative welfare states (e.g., hunger, discomfort, pain, and distress). Good animal welfare also requires facilitating positive welfare states [5,27]. Examples include species-specific rewarding behaviours (e.g., foraging, play, nest-building) and affiliative animal–human interactions.

In addition, international development organisations should apply these general principles:If production systems do not meet animals’ welfare needs, they will be discouraged in ongoing projects and not introduced in new projects.There should be safe and suitable animal facilities and appropriately trained individuals in place before the animals arrive.In case welfare issues arise, there should be written contingency plans to minimise animal suffering.An appropriate welfare assessment tool should be in place, for example, Welfare Quality^®^ protocols (although WQ may not be appropriate outside Europe).

### 3.2. Governance Structure for Institutionalising the Guidelines

A clear governance structure will oversee the guidelines’ implementation (Figure 1). The Steering Group will have overall responsibility for ensuring the guidelines are followed; the advisors will offer specialist expertise; and individual task forces will address specific welfare issues. Their roles are outlined in more detail below.

#### 3.2.1. Steering Group

The Steering Group will have overall institutional responsibility for implementing these guidelines. It will give the international development organisation strategic direction and guidance on animal welfare, as well as track progress and address any issues. The Steering Group will oversee task forces, which will ensure compliance “on the ground”. To achieve best practice, the Steering Group could also collaborate with other organisations (e.g., governments and non-governmental organisations) to promote animal welfare at a regional and global level. Steering Group members will be drawn from all levels of the international development organisations, from senior management to project workers.

#### 3.2.2. Advisors

Advisors will be external individuals, who offer guidance and monitor animal welfare within the international development organisations. They may focus attention on specific priorities; provide expert opinion and recommend courses of action; accelerate, monitor, and evaluate progress; and encourage international development organisations to make specific commitments. Advisors may either represent major donors and funding agencies, or be experts in animal welfare and other relevant topics.

#### 3.2.3. Task Forces

Task forces will be responsible for ensuring these guidelines are followed “on the ground”. The Steering Group will form them depending on organisational need. For example, task forces may cover working animals, animals given to farms or households, research animals, transport and slaughter, or in-service and pre-service training for animal health professionals. These task forces will advance Steering Group-determined priorities, objectives, and measures of progress and impact, as well as feeding back any issues to the Steering Group. They will primarily consist of project workers and other staff who are in day-to-day contact with animals.

### 3.3. Actions

#### 3.3.1. Staff Training

All project staff will receive sufficient training to competently implement these guidelines.Depending on the individual’s responsibilities, their training may cover animal sentience, suffering, needs, and interests; the Five Freedoms; welfare-friendly animal handling, transport, and slaughter; nutrition; waste management, drainage, and disposal; reproductive management techniques; biosecurity; basic clinical examination, indicators of disease and poor welfare, and pain management; and animal welfare legislation.Staff will complete their training before being assigned responsibility for any animals. Thereafter, refresher sessions will be provided when necessary.Training will not cause animals unnecessary pain, distress, or harm.

#### 3.3.2. Standard Operating Procedures

5.The animal carers/managers will write standard operating procedures (SOPs) for the care and management of every animal species.6.SOPs will describe day-to-day husbandry, including feeding, watering, cleaning, health-checks, handling, restraint, transport, and slaughter.7.SOPs will outline measures to prevent negative welfare outcomes (such as wounds, lameness, pain, and disease), promote positive welfare outcomes (such as enrichment and humane handling), and name the staff responsible for adherence.8.SOPs should be based on existing welfare-relevant literature (a list of resources is available on the Welttierschutzstiftung website [28]).9.The project’s manager and animal carers are responsible for ensuring adherence to SOPs.10.SOPs will be reviewed annually and updated when necessary. *A sample of the SOP format is provided in Appendix A, although this is a template only.*

#### 3.3.3. Water

11.All animals will receive an adequate supply of safe (“potable”) drinking water, with appropriate facilities for its storage and distribution.12.The water supply will be adequate for each animal’s species, age, and physiological needs.13.Animals will be prevented from consuming contaminated water or other contaminants likely to cause disease.14.Water troughs will be shaded to prevent over-heating.

#### 3.3.4. Food

15.Forage, feed, and fodder will be of appropriate quantity and quality.16.Feed will not present a serious risk of transferring, directly or indirectly, infectious agents, pesticide residues, or other toxins.17.Feed will be adequate for each animal’s species, age, and physiological needs.

#### 3.3.5. Housing

18.Housing will be comfortable for animals and not adversely affect their physical or mental health.19.Housing will be kept clean and maintained in a way that does not present a serious health risk to the animals, owners/carers, or the community.20.Housing will provide adequate ventilation and other features that enable animals to regulate their temperature (e.g., by accessing shade).21.Housing will provide sufficient space for the animals to express normal behaviours, species-appropriate shelter, and visual barriers, and will not be stocked to such an extent that normal behaviours cannot be performed.22.The animals will be kept in an appropriate perceptual environment for their species (i.e., minimising aversive noise, light, etc.).23.Whenever practical, animals will have outdoor access.

#### 3.3.6. Social Isolation

24.Animals will be kept in appropriate social conditions, so they can express normal social behaviours.25.Animals will not be socially isolated, except for veterinary, animal welfare, or worker safety reasons. Where the separation is long term and not urgent, a qualified veterinarian must approve it.

#### 3.3.7. Enrichment

26.Animals will receive regular enrichment to facilitate natural behaviour expression.

#### 3.3.8. Drainage and Waste Disposal

27.Adequate drainage and waste disposal facilities will be provided. They will be designed and constructed to ensure infection prevention and control. For example, where necessary, drainage will end in a soak pit; litter and refuse will be deposited in a designated pit; and compounds/yards will be kept hygienic through daily cleaning and monthly disinfection with calcium hydroxide or another disinfectant (ensuring that any chemicals used are safe and not harmful to the animals).

#### 3.3.9. Disease

28.Wherever possible, animal health will be maintained.29.Necessary vaccination, deworming, and parasite control will be given at the appropriate time. This should always be under veterinary advice/supervision.30.To the greatest extent practical, infection will be prevented and controlled. For example, infected animals or herds will be isolated; pathogen occurrence will be monitored for early warnings of disease; and sick animals will be promptly treated.31.The animal carers/Project Manager (preferably a certified vet) will give animals a basic clinical checkup daily and a comprehensive clinical checkup weekly.32.Whenever necessary, animals will be given effective pain management (including analgesics and/or anaesthetics), including when injured or undergoing surgery, or for painful management procedures.33.Euthanasia will be considered for health conditions where the animal is in pain with no chance of full recovery.34.Antibiotics will only be used to treat a disease diagnosed by a certified vet, and based on current antimicrobial stewardship guidelines.

#### 3.3.10. Invasive Procedures

35.Project Managers/animal carers will not practice, or will actively work to reduce, unnecessary painful management procedures.36.Where invasive procedures are necessary (e.g., for worker safety or animal welfare reasons), the animals will be given effective pain relief.

#### 3.3.11. Transport

37.Where possible, knowledge of flight distance will be used to move the animals. The flight zone is the area surrounding the animal which, if entered by a potential predator (including humans), will cause the animal to move away.38.Transport times will be minimised and live animals will not be transported for longer than 8 h, unless approved by a certified veterinarian. Where longer transport times are inevitable, the animals must have a rest stop with sufficient food, water, and time to rest.39.Animals likely to be aggressive to each other (e.g., mature bulls) will not be mixed during transportation.40.There will be enough personnel who are patient, considerate, competent, and familiar with species-specific transport requirements and behaviour. These personnel will not use electric prods or excessive physical force (i.e., force that causes pain or injury).41.To minimise disease transmission, trucks will be disinfected between each group of animals transported.

#### 3.3.12. Slaughter

42.Before slaughter, all livestock will be effectively stunned. For mammals, this will involve a bolt-gun aimed at the top of the head. For poultry, electrical stunning will be used.43.Livestock will have their throats cut using a sharp knife, ensuring that the key blood vessels are cut (carotid arteries and jugular veins).44.Everyone involved in the slaughter will be competent and properly trained.

#### 3.3.13. Breeds

45.All animals acquired will be an appropriate breed for the environment where they will be kept.

#### 3.3.14. Record-Keeping

46.Comprehensive records will be kept for all animals. These will include veterinary care and treatments, transport, and slaughter.

#### 3.3.15. Monitoring and Evaluation

47.The relevant Project Manager and/or task-force will perform a quarterly audit of all animals and animal facilities. Where welfare issues are identified, these should be promptly addressed.

## 4. Conclusions

In consultation with international development organisations operating in the Global South, their donors, and recipients, we developed the first comprehensive animal welfare guidelines for their projects and activities. These guidelines cover institutional governance, staff training, standard operating procedures, water, food, housing, social isolation, enrichment, drainage and waste disposal, disease, invasive procedures, transport, slaughter, breeds, record-keeping, and monitoring and evaluation procedures. If our guidelines are institutionalised, they should ensure good animal welfare standards, without being too demanding.

We now hope that international development organisations in the Global South commit to the guidelines and implement them. We will disseminate the guidelines through webinars, conferences, and online resources. However, it is crucial that IDOs go beyond committing to the guidelines and actually implement them “on the ground”. This may not happen overnight. We must be flexible and understand that these organisations have many competing priorities. In the long run, however, good welfare standards will benefit international development organisations, donors, recipients, and the animals themselves.

## Figures and Tables

**Figure 1 animals-14-02012-f001:**
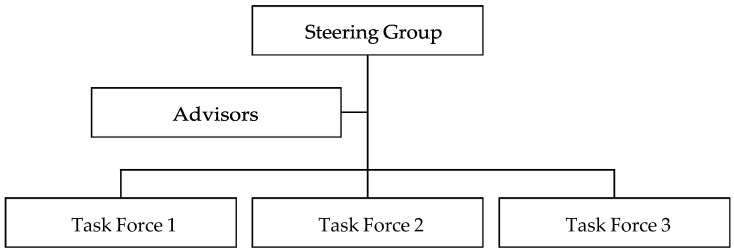
Proposed organisational governance structure for institutionalising the animal welfare guidelines.

## Data Availability

The original contributions presented in the study are included in the article/Appendix A; further inquiries can be directed to the corresponding author.

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
