# Peer review of "Animal Welfare Guidelines for International Development Organisations in the Global South"

_animals, 2024, doi:10.3390/ani14132012_

Round 1

Reviewer 1 Report

Comments and Suggestions for Authors

The construct of this submission appears to be:

- “For most international development organisations, animal welfare is simply not “on the radar” 

- “Effective and transparent animal welfare policies are, therefore, crucial for international development organisations to keep their social licence, both among beneficiaries and donors“

- “To our knowledge, however, there is no set of animal welfare guidelines tailored for international development organisations”

- “Here, we develop animal welfare guidelines for international development organisations operating in the Global South, especially Africa. We first outline our methodology, including extensive stakeholder engagement.”

  • “Our aim is for international development organisations to adopt and institutionalise these guidelines, so they actively foster good animal welfare practice.”

In reviewing this approach, there appear to be some outstanding questions.

1. The broad statement that animal welfare is not on the radar for most development organisations needs further attention. 

World Bank: https://documents1.worldbank.org/curated/en/958081468320947271/pdf/938420WP0Box3800Animal0Welfare02014.pdf

UN and wider: https://www.wellbeingintlstudiesrepository.org/cgi/viewcontent.cgi?article=1001&context=eth_leg

https://www.unep.org/resources/perspective-series/issue-no-34-why-animal-welfare-important-sustainable-consumption-and 

There are organisations that have expertise in this area is development and animal welfare:

https://www.thebrooke.org/our-work/working-equids-sustainable-development 

There is also discordance of assumptions within the submission, for example later it is stated “The disparity between survey responses and project visits highlighted that international development organisations may have animal welfare policies, but these are not apparent on their projects”.

 > Therefore a much deeper review and critique of the primary assertion of “not on the radar” is needed. Is the existing situation with IGOs and NGOs that of mission statements with no policies, policies but not implementation, of failure to implement these policies?

2. The survey and visits were limited in scope, and there are understandable issues with the confidentiality required to elicit accurate response in the context of maintaining relationships with funders. 

> However precisely because of the small sample size much more detail and transparency is needed. For example the claim for SOPs being in place but not used could indicate that failure of delivery, rather than the policies themselves, are part of the issue.

3. The nature of the IGO and NGO landscape means that they tend to reflect each others, and wider, standards. This can be both a block to innovation but also a route to change. This submission reports that the view of the WOAH was “overwhelmingly negative” and standards were “too general, too complex, and did not reflect on the different situations in different countries”. 

However as a starting point WOAH principles are similar to those posited in this submission:

https://www.woah.org/en/what-we-do/standards/codes-and-manuals/terrestrial-code-online-access/?id=169&L=1&htmfile=chapitre_aw_introduction.htm

> Again a webinar is reported, but without any further detail, so this should be provided especially as to why the top tier of WOAH guidelines could not be developed to produce WOAH standards that are useful and also sellable to IGOs and NGO’s?

4. There is a claim to have achieved consensus  support for these guidelines. It is not clear from the conference report, https://www.aawconference.org/2022/6th_Africa_Animal_Welfare_Conference_Action_2022_Report.pdf  whether the resolution to

“Request AU Member States through AU-IBAR and development partners to adopt Guidelines for Incorporation of Animal Welfare in Development Cooperation Projects.” https://www.aawconference.org/index.php/resources/past-conferences  is related to these specific guideline in this submission of the various guidelines mentioned in the conference report, especially those of WOAH.

> This issue must be specifically addressed.

5. The aim of ensuring good delivery of animal welfare via IGO and NGO is laudable and benefits them as well as people, animals and the wider environment. This is also a useful bottom-up approach to highlighting gaps in delivery.

This could well be another example of the challenge in implementing the “localisation agenda” https://devex.shorthandstories.com/the-localization-agenda-2-0/index.html

So the significant question is how IGOs and NGO’s can be influenced to adopted and ensure delivery, as the reality is that funding sources drive this, i.e does this in reality have, at present, to be a top-down activity to achieve success.

> A fundamental question is why an approach excluding working through WOAH in particular was  taken, and in turn how any guideline, whatever their origin, can actually be delivered on the ground?

Overall, I think a fundamental concern with this submission is if IGOs and NGOs are actually ignoring animal welfare, how in the current funding environment will suggesting these SOPs  actually achieve the desired outcome.

I could suggest that a stronger submission might to to expand on the detail of the activities that get the authors to the point that  animal welfare is an issue but policies and/or delivery are sub-optimal, and then propose options to address this gap, in particular how funders can be influenced to improve matters. 

At present I an unsure how this submission, terms of real-world outcomes, will add value.

As an important aside I do not think the example of the SOP in supplementary materials is in any way useful, it is just headings.

As a further  aside, in general I prefer Univeristy emails to be used when such an affiliation is recorded by an author: [email protected]

Author Response

The comments together with the responses are in the word document attached, thanks

Reviewer 2 Report

Comments and Suggestions for Authors

Overall this is an interesting paper, with clear relevance to the sector. It is designed to have impact in the work of international development organisations, and has been developed in clear partnership. It is largely well written and communicates clearly. At times content is a little vague, and further explanation/provision of examples would be beneficial to ensure the reader is on the same page as the authors. I have identified some more detailed points for your consideration below, and hope these are useful in developing the script. I have identified a few typos, but have by no means proof read extensively.  

L42-43 - I wonder if mentioning neglect in relation to animal welfare in the context of the organisations may be inflammatory, with neglect being a form of animal cruelty - perhaps less contentious language may be beneficial here, such as "IDOs have failed to consider...". 

L43 - what is the citation here? (212) - assume typo. 

L61-66 - Is continuous housing being promoted in the Global South by these organisations? Admittedly it is not ideal accommodation for livestock welfare, but is it expected that the organisations mentioned are promoting such practices? I wonder if more examples whereby animals are directly suffering, such as in aid / natural disaster situations due to lack/poor advice could be added to illustrate the problem more clearly/broadly? It is in these times of crisis where I expect greatest animal suffering could take place, but this doesn't currently come through in this introduction, or in the rest of the article. If this situation is beyond the scope of your work to date it might be useful to note this in the article, or refer the reader to other sources.  

L78-79 - Can an example of how donors allocate funding based on animal welfare be provided to improve the clarity of this point? A little more detail throughout this section re: the work these organisations are doing would be useful. It is easy to assume that most of the time these organisations are working in areas in time of crisis, so if this is not solely to context under which you see the guidelines being used it would be useful to explore this for the reader throughout this chapter. 

L89-90 - "We first outline our methodology, including extensive stakeholder engagement. Then, we outline the guidelines themselves." - This is superfluous in my opinion. 

L106 - Welttierschutzstiftung - it would be useful to know who this organisation is/why they are the authority on these organisation or how this list was compiled - e.g. projects they have worked on over X no. of yrs. 

L108 - Errant ) by 117

L111-114 - The scale of these projects would be important to report here - how many animals/personnel are included in each?

Throughout - abbreviate international development organisations to IDOs for ease of reading

L160 - worth mentioning donkeys specifically?

L190 - Some explanation of Figure 1 is needed before it is presented in the text for clarity of its relevance

Conclusion - some consideration of how you intend to disseminate these guidelines to IDOs would be beneficial here, giving insight into their application and dissemination. 

References - some issues with incomplete references e.g. #3

Comments on the Quality of English Language

The article is largely clearly written. Use of language is appropriate. 

Author Response

(The authors gave the same response as above.)

Reviewer 3 Report

Comments and Suggestions for Authors

The manuscript "Animal welfare guidelines for international development organisations" aims to highlight the often-neglected issue of animal welfare within international development activities. Its strength lies in providing principles for good animal welfare, covering areas like food, housing, veterinary care, transport, and slaughter etc. Notably, the guidelines are based on experiences from Africa, ensuring relevance to the region. Adoption of these guidelines can improve animal welfare standards and enhance the reputation of development organizations.

General comments:

The manuscript addresses a highly relevant and actual topic with significant international importance. Currently, the issue of animal welfare in development projects lacks sufficiently detailed scientific literature, making this contribution particularly valuable. The manuscript is impressive in its scope, involving numerous organizations and experts in the creation of the guidelines. Although the guidelines primarily draw from experiences in Africa, or as the Authors term it, the Global South, emphasizing this connection could enhance the manuscript's specificity without detracting its value.

The guidelines might indeed be applicable in the Global North, but the manuscript does not provide detailed evidences and support for this. Therefore, it is suggested to either include evidence supporting their applicability in different contexts or focus on their primary relevance to the Global South.

While the references used are appropriate, some claims lack adequate citations and appear more as opinions. These should be supported by relevant literature to enhance their credibility. Additionally, the "Materials and Methods" section contains results, indicating a need to reconsider the structural division of the manuscript.

Furthermore, within the guidelines section, it is unclear what constitutes the transitional text and what are the guidelines themselves.

Specific comments:

Some sections of the article (for example lines 64-66) contain statements that seem more like personal opinions rather than well-supported arguments. These claims lack adequate references to scholarly literature, which undermines their validity. In a scientific work, it is important to avoid using subjective expressions such as "the right thing to do" (line 74). Such phrases are not suitable for academic writing as they can be perceived as opinion-based rather than evidence-based.

Several key concepts in the article are not adequately explained. For instance, the criteria used to categorize countries as "poor" are not specified (line 75). It is unclear whether this classification is based on GDP, other economic indicators, or a combination of factors. Additionally, the term "social license" mentioned in line 80 is not explained.

When using abbreviations for the first time, it is essential to provide explanations for their meanings to ensure clarity for the readers. For instance, when "S. P." is mentioned for the first time in line 112, it has to be explained. Similarly, the abbreviation "IGO" in line 117 requires clarification to indicate what it represents.

In the Abstract it is mentioned that the manuscript is based on experiences from Africa, but this crucial point is not highlighted in the Simple Summary. It is important to emphasize that the manuscript focuses on research in Africa, even though it is revealed in line 154 that organizations from Europe and the USA were also involved in its development.

There are certain sections in the manuscript where more precise data would be necessary for a scientifically rigorous document. For instance, terms like "several rounds of feedback" (line 145) lack specificity. Additionally, phrases such as "from over 30 African countries' Ministries of Agriculture" (line 149-150) and "almost all African NGOs" (line 150) would benefit from precise percentages or numbers to convey a clearer picture of the extent of involvement.

(The article's references are certainly not formatted correctly, the in-text citations are presented as footnotes.)

Author Response

(The authors gave the same response as above.)

Reviewer 4 Report

Comments and Suggestions for Authors

This is a well needed and well written paper - I am aware of certain INGOS eg Oxfam, Send a Cow which have used animal welfare guidelines for around 20 years but there is a need for standardisation and audit of these guidelines. 

Author Response

(The authors gave the same response as above.)

Round 2

Reviewer 1 Report

Comments and Suggestions for Authors

This is much improved in revision. There are a number of typographical issues within the editing, their meaning is clear but they need attention.

Comments on the Quality of English Language

See above